# Generation of Cochlear Hair Cells from Sox2 Positive Supporting Cells via DNA Demethylation

**DOI:** 10.3390/ijms21228649

**Published:** 2020-11-17

**Authors:** Xin Deng, Zhengqing Hu

**Affiliations:** 1Department of Otolaryngology-Head and Neck Surgery, Wayne State University, Detroit, MI 48201, USA; xdeng17@hawk.iit.edu; 2John D. Dingell VA Medical Center, Detroit, MI 48201, USA

**Keywords:** 5-azacytidine, cell conversion, epigenetics, hair cells, regeneration, supporting cells

## Abstract

Regeneration of auditory hair cells in adult mammals is challenging. It is also difficult to track the sources of regenerated hair cells, especially in vivo. Previous paper found newly generated hair cells in deafened mouse by injecting a DNA methyltransferase inhibitor 5-azacytidine into the inner ear. This paper aims to investigate the cell sources of new hair cells. Transgenic mice with enhanced green fluorescent protein (EGFP) expression controlled by the Sox2 gene were used in the study. A combination of kanamycin and furosemide was applied to deafen adult mice, which received 4 mM 5-azacytidine injection into the inner ear three days later. Mice were followed for 3, 5, 7 and 14 days after surgery to track hair cell regeneration. Immunostaining of Myosin VIIa and EGFP signals were used to track the fate of Sox2-expressing supporting cells. The results show that (i) expression of EGFP in the transgenic mice colocalized the supporting cells in the organ of Corti, and (ii) the cell source of regenerated hair cells following 5-azacytidine treatment may be supporting cells during 5–7 days post 5-azacytidine injection. In conclusion, 5-azacytidine may promote the conversion of supporting cells to hair cells in chemically deafened adult mice.

## 1. Introduction

In the mammalian organ of Corti, inner hair cells are arranged in a single row and form synapses with afferent nerve terminals of spiral ganglion neurons [1]. The three rows of outer hair cells detect acoustic information and amplify the signals [1]. Supporting cells separate hair cells and provide support for the deflection of hair cells [2]. Hair cells and supporting cells are arranged in a mosaic pattern, in which each hair cell is separated from the next by an interceding supporting cell [3]. The hair cells and supporting cells originate from common progenitors [4,5]. In the developing mammalian cochlea, Sox2 is required for the formation of prosensory area [6,7]. Development of hair cells is accompanied by downregulation of Sox2 [8]. In mature adults, Sox2 expression is restricted to supporting cells, while hair cells no longer express Sox2 [8].

In nonmammalian vertebrates such as avian, fish and amphibians, regeneration of damaged hair cells can be achieved through mitosis of supporting cells or transdifferentiation of supporting cells [9,10,11]. Unlike nonmammalian vertebrates, deficits in adult auditory hair cells will lead to permanent hearing loss in mammals [1,3,12]. It is challenging to regenerate auditory hair cells in damaged adult mouse cochlea. In mammals, proliferation and regeneration of new hair cells from supporting cells have been observed in neonatal mice [13,14]. It was reported that a subset of Lgr5+ supporting cells collected from postnatal mice could self-renew and give rise to hair cells [13]. Generation of hair cells from supporting cells in the cochlea of adult mice remains largely uninvestigated [14].

To regenerate hair cells in adult mammalian cochlea, epigenetic modification may be a promising method [15,16,17]. Epigenetic regulation can modify the genomic structures and activities in response to intracellular and environmental cues, resulting in expression of specific genes, thereby mimicking the normal developmental processes [16,17,18]. Epigenetic mechanisms include histone modification and DNA methylation [18]. DNA methylation is mediated by DNA methyltransferase 1 (DNMT1), DNMT3A and DNMT3B, which is usually associated with silencing of gene expression [18]. DNA methylation is a reversible process and the process can be maintained throughout life [19]. For example, 5-azacytidine (5-aza) can inhibit DNA methyltransferases (DNMTs), promote DNA demethylation and induce expression of silenced genes [16,17,18,19]. Our previous data indicate the role of 5-aza in stimulating the differentiation of hair cell-like cells in cultured inner ear stem cells in vitro [17]. Recently, regenerated hair cells are observed in the inner ear of deafened adult mice following application of 5-aza to deafened adult mice in vivo [20]. New hair cells are found to express several hair cell specific markers including Myosin VIIa, Myosin VI and Pou4f3, while the control group receiving saline injection lacks hair cell regeneration [20]. The level of DNMT1 mRNA is reduced following 5-aza treatment, supporting the role of 5-aza as a DNMT1 inhibitor in regeneration of hair cells [20]. These data suggest the role of 5-aza in regenerating hair cells through DNA demethylation [20]. The sources of the regenerated hair cells, however, remain to be determined.

In this research, the sources of regenerated hair cells by 5-aza was explored in the deafened mouse model. This study hypothesizes that the sources of regenerated hair cells include the Sox2 positive supporting cells. Sox2 is selected to prelabel all supporting cells because Sox2 is known to express in all types of supporting cells in the mature murine inner ear [8]. Therefore, a strain of transgenic mice that the expression of enhanced green fluorescent protein (EGFP) is expressed under control of the endogenous Sox2 regulatory element (Sox2-EGFP transgenic mice) was applied to test the hypothesis by observing the fate of Sox2 positive supporting cells. The results of this study may promote development of novel treatment for hair cell regeneration in the future.

## 2. Materials and Methods

### 2.1. Animals

Young adult (4–6 weeks old, either sex) Sox2-EGFP transgenic mice (Jax Mice) were applied in the study. The care and use of animals was approved by local Institutional Animal Care and Use Committee (IACUC). The animal study procedures were carried out in accordance with the approved guidelines.

### 2.2. Deafening

Kanamycin (1 g/kg body weight, s.c.) was administered followed by furosemide injection (300 mg/kg body weight, i.p.) after 25–30 min. Pure tone baseline auditory brainstem responses (ABR) was measured to evaluate the hearing function of mice at frequencies of 8 kHz, 16 kHz and 24 kHz with stimulation of 10–90 dB SPL in 5–10 dB steps using the RP2.1 and RZ6 system (Tucker-Davis Technology; TDT). The TDT System 3 software-controlled signal generation and response collection. ABR evaluations were performed on the same mice before deafening to record the baseline, and three days post-deafening to determine the effects of deafening. Mice showing hearing threshold shift >40 dB SPL were included for the following studies. In this study, all the mice were successfully deafened.

### 2.3. Inner Ear Surgery

As reported in the previous study [20], 4 mM of 5-aza was injected into 32 deafened mouse inner ears. Under deep anesthesia, the left temporal bone was exposed and opened to visualize the basal cochlear turn and round window. By inserting a small catheter into the round window, 0.2 µL of 5-aza solution was delivered into the inner ear via a microliter syringe (Hamilton, Reno, NV, USA). After surgery, 5-aza group of mice were followed for 3, 5, 7 and 14 days (n = 8 mice in each group) to observe the fate of hair cells and supporting cells. The control group (n = 8 mice) was handled the same as the 5-aza group, except for that saline (vehicle solution used to dissolve 5-aza) was delivered into the inner ear instead of 5-aza. The control group of mice was followed for 14 days.

### 2.4. Histology and Immunofluorescence

Mice from both 5-aza and control groups were euthanized at the end of each follow-up period, and cochlear samples were harvested and fixed by 4% paraformaldehyde (Sigma, St. Louis, MO, USA). Ethylenediaminetetraacetic acid (EDTA, 0.1 M, Sigma, St. Louis, MO, USA) was used to decalcify cochleae for 5–7 days. Cochlear samples were collected by cryosectioning at 10 µm thickness. Here, cochlear section instead of wholemount preparation was chosen because in the previous publication using the wholemount preparation, newly generated hair cells and supporting cells were not in the same surface [20]. While some of the supporting cells can be observed with the hair cells, most supporting cells were beneath the surface hair cells [20], making it challenging to detect the hair cells and supporting cells simultaneously. Thus, cochlear sections that provide side views of the organ of Corti were applied in this study to detect whether new hair cells are generated from supporting cells. For immunostaining of sections from both 5-aza and control groups, cochlear sections were treated with phosphate buffered saline (PBS) containing 5% normal donkey serum (Jackson ImmunoResearch, West Grove, PA, USA) and 0.2% Triton X-100 (Sigma, St. Louis, MO, USA) for 30 min. Samples were incubated in primary antibodies at 4 °C overnight, followed by corresponding secondary antibody incubation at room temperature for 1–2 h. The primary antibodies used in this study were: anti-Myosin VIIa (1:100; Developmental Studies Hybridoma Bank, DSHB, Iowa City, IA, USA and Proteus Bioscience, Ramona, CA, USA) and anti-Sox2 (1:200; R&D system, Minneapolis, MN, USA). Secondary antibodies included Cy3 and Dylight-647 conjugated antibodies (1:500; Jackson ImmunoResearch, West Grove, PA, USA). 4,6-Diamidino-2-Phenylindole (DAPI; Invitrogen, Carlsbad, CA, USA) was used to label all nuclei. Leica SPE confocal microscopy was used to capture all the images.

### 2.5. Data Analysis

After immunostaining, the numbers of cells with Myosin VIIa signals only, Sox2-EGFP signals only and double labeled cells on the organ of Corti were counted. Four animals (n = 4) from 3-, 5-, 7- and 14-day post-surgery groups were included in the quantification. The percentages of Sox2-EGFP positive cells, Myosin VIIa positive cells and double-labeled cells were calculated as [(number of Sox2−EGFP positive cells)/(number of Sox2−EGFP positive cells+number of Myosin VIIa positive cells)]·100%, [(number of Myosin VIIa positive cells)/(number of Sox2−EGFP positive cells+number of Myosin VIIa positive cells)]·100% and [(number of double−labeled cells)/(number of Sox2−EGFP positive cells+number of Myosin VIIa positive cells)]·100%, respectively. For statistics, the Kruskal–Wallis test was performed using SigmaPlot to test the significance among the four groups (3-day, 5-day, 7-day and 14-days). If there was a significant difference, Student–Newman–Keuls post-hoc comparisons were used to identify the sources of significance.

## 3. Results

### 3.1. Sox2 and GFP Were Coexpressed at the Organ of Corti in Sox2-EGFP Transgenic Mice

Before deafening and treatment to the mice, the expression of EGFP and Sox2 was examined in adult Sox2-EGFP transgenic mice. Bright field image showed the structure of the cochlea and organ of Corti (Figure 1). Low magnification figures showed that Sox2, Myosin VIIa and EGFP mainly presented at the organ of Corti region (Figure 1). High magnification figures showed that within the same cell, expression of Sox2 protein was concentrated in the nucleus while expression of EGFP was in the cytoplasm (arrows in Figure 1). Immunolabeling the cochlear section with hair cell marker Myosin VIIa indicated the location of hair cells (arrowheads in Figure 1). Merged figures suggested that the labeling of hair cells and Sox2 positive supporting cells were separated (arrows and arrowheads in Figure 1). Sox2 and EGFP were expressed in the same cells (arrows in Figure 1). These data showed the expression of Sox2 in the cochlea of the Sox2-EGFP transgenic mouse, and most importantly, EGFP expression can represent the presence of Sox2 positive supporting cells.

### 3.2. 5-aza May Stimulate Regeneration of Hair Cells from Sox2 Positive Supporting Cells in Deafened Mice

New hair cell generation may involve complicated cellular mechanisms, in which cell cycle reentry and/or direct cell type conversion may play important roles. Our previous data showed that mitosis was not observed up to two weeks after 5-aza injection [20], suggesting that new hair cells may not be generated via cell cycle reentry. The other possibility for new hair cell generation is direct cell type conversion, in which supporting cells may directly convert into hair cells. Sox2 is expressed in the developing inner ear and is specifically expressed in all subtypes of supporting cells of the mature cochlea [3,21,22]. To characterize whether Sox2-expressing supporting cells converted into new hair cells, Sox2-EGFP transgenic mice were treated identically to the previous research that utilized the wildtype mice: mice were deafened and treated with 5-aza 3 days after deafening [20]. Observation of the mouse started at earlier stages after treatment to capture any potential transition of supporting cells to hair cells. Three days following 5-aza treatment, Sox2-EGFP-expressing supporting cells were observed in the organ of Corti (Figure 2A). No Myosin VIIa positive cell with hair cell morphology was observed at this time point (Figure 2A). During days 3–5 post treatment, expression of Sox2-EGFP and Myosin VIIa was observed (Figure 2A). Specifically, cells expressing Sox2-EGFP, Myosin VIIa and both were detected in the organ of Corti region (Figure 2A). EGFP and Myosin VIIa double-labeled cells were observed during 5–7 days following 5-aza treatment (Figure 2A). During 7–14 days after surgery, cells in the organ of Corti expressed only Myosin VIIa or only EGFP (Figure 2A). No EGFP and Myosin VIIa double-labeled cells was detected during 7–14 days after treatment (Figure 2A). The Myosin VIIa-expressing cells showed columnar apical-basolateral polarity, which is similar to native hair cells. These data suggest that Sox2-expressing supporting cells may directly convert into new hair cells without mitosis.

Quantitative study was performed to further explore the source of regenerated hair cells. The numbers of Sox2-EGFP-expressing cells and Myosin VIIa positive cells were counted using animals of 3 days, 5 days, 7 days and 14 days post-surgery. The percentages of Sox2-EGFP positive, Myosin VIIa positive and double labeled cells were calculated. The results show that three days post-surgery, all cells were Sox2-EGFP positive, whereas Myosin VIIa positive hair cells were not observed, indicating the transition from supporting cells to hair cells has not begun yet (Figure 2B). Five days post-surgery, there were 55.78% ± 4.56% Sox2 positive cells, 44.22% ± 4.56% Myosin VIIa positive cells and 31.62% ± 7.05% double-labeled cells in the organ of Corti area (Figure 2B). These data reveal that during the early stage of hair cell regeneration, a proportion of the cells at the organ of Corti were both Myosin VIIa and Sox2-EGFP positive, suggesting beginning of the transition from supporting cells to hair cells. Seven days after surgery, the percentages of Sox2-EGFP positive, Myosin VIIa positive and double-labeled cells were 54.58% ± 5.58%, 45.42% ± 5.58% and 8.33% + 6.31%, respectively (Figure 2B), indicating the transition from supporting cells to hair cells was in progress. As far as 14 days after surgery, the percentages of Sox2-EGFP positive cells and Myosin VIIa positive hair cells were 57.59% ± 6.96% and 42.41% ± 6.96%, respectively (Figure 2B). Double-labeled cells were not observed (Figure 2B), suggesting the transition of hair cells may have been completed by two weeks following 5-aza injection. Statistical analysis revealed that the percentages of Sox2-EGFP positive cells were significantly different from 3-day to 14-day post-surgery (n = 4, *p* = < 0.001). Student-Newman–Keuls comparison showed that the percentage of Sox2-positive cells was significantly higher in the 3-day post-5-aza treatment compared to the 5-day, 7-day and 14-day post-surgery groups (3-day vs. 5-day, *p* < 0.001; 3-day vs. 7-day: *p* < 0.001; 3-day vs. 14-day, *p* < 0.001). Similar results were observed in the comparison between the percentages of Myosin VIIa positive cells. There were significant differences among the percentages of Myosin VIIa positive cells from 3-day to 14-day post-surgery (n = 4, *p* = < 0.001). The percentages of Myosin VIIa positive cells was significantly smaller in the 3-day post-5-aza treatment compared to the 5-day, 7-day and 14-day post-surgery groups (3-day vs. 5-day, *p* < 0.001; 3-day vs. 7-day: *p* < 0.001; 3-day vs. 14-day, *p* < 0.001). For the double-labeled cells, there was significant difference among the four groups (n = 4, *p* < 0.01). The percentage of double-labeled cells was significantly higher in the 5-day post-surgery compared to the other groups (5-day vs. 3-day, *p* < 0.05; 5-day vs. 7-day: *p* < 0.05; 5-day vs. 14-day, *p* < 0.05). Taken together, these data suggest that the transition of hair cells from supporting cells may be during 5 days to 7 days following 5-aza injection.

As consistent with previous data [20], no regenerated hair cells were observed in the control group of deafened mice that were treated with saline injection. At the organ of Corti, only Sox2 and EGFP signals were detected two weeks after saline injection (Figure 3), suggesting that Myosin VIIa-expressing hair cells were not observed in deafened mice of the saline treatment group.

## 4. Discussion

In this study, the DNMT1 inhibitor 5-aza was injected into chemically deafened mature mouse cochleae. Using Sox2-EGFP transgenic mice, regeneration of hair cells was observed. Overlapping expression of Sox2 and Myosin VIIa was observed as early as 5-day post 5-aza injection and was not observed two weeks post-surgery. Quantitative study indicating possible transition from supporting cells to hair cells during 5–7 days following 5-aza treatment. These data suggest that supporting cells may be the cell source of new hair cell generation after 5-aza treatment.

Before investigating the cell sources of regenerated hair cells by 5-aza, the expression of Sox2 in the cochlea was examined. The low magnification images of the cochlear sections indicate the expression of Sox2 in the cochlea (Figure 1). Consistent with other studies, the locations of Sox2 positive cells mainly focused on the organ of Corti, stria and spiral ganglion (Figure 1) [23,24,25].

The supporting cells and hair cells at the organ of Corti were indicated by anti-Sox2 and anti-Myosin VIIa antibodies, respectively. Immunostaining showed coexpression of Sox2 and EGFP within the same cells in the organ of Corti (Figure 1). In addition, Myosin VIIa is expressed in hair cells, while EGFP was observed in the neighboring supporting cells (Figure 1). The EGFP positive cells were either underneath or adjacent to the Myosin VIIa positive hair cells (Figure 1). These data support the role of EGFP signals as the presence of supporting cells at the organ of Corti.

Previous published study evaluated the effects of 5-aza in terms of chemotherapy. In the culture of mouse utricle epithelia-derived progenitor cells, treatment of 8 µM of 5-aza for 72 h led to significantly decreased number of viable cells, while lower concentrations did not cause decreased number of viable cells, indicating that high concentration of 5-aza may be toxic to mouse utricle epithelia-derived progenitor cells [16]. In the in vivo study, 4 mM 5-aza (n = 8) was injected into the mouse inner ear. No inflammation or other significant side effect was observed in all injected mice for up to six weeks following injection [20]. Therefore, 5-aza is likely to be safe if used in a proper concentration in the future application as a chemotherapy.

Our previous study shows that at 7-day post 5-aza injection, hair cells were observed in the damaged cochlea. Further, the number of regenerated hair cells was increasing from 7-day to 14-day post 5-aza injection [20]. Therefore, the generation of hair cells may happen within the first two weeks after 5-aza injection. Here, 3-day, 5-day, 7-day and 14-day post 5-aza treatment were selected to track the transition of hair cells from supporting cells. Immunostaining of Myosin VIIa and Sox2-EGFP showed the fate of hair cells and supporting cells. Coexpression of Myosin VIIa and Sox2-EGFP was observed in the sections of 5-day and 7-day post-5-aza treatment groups, indicating the regeneration of hair cells from supporting cells was in progress during 5–7 days post 5-aza injection. Expression of Sox2-EGFP and Myosin VIIa was separated at 14-day post-5-aza treatment, suggesting the transition from supporting cells to hair cells may have been accomplished. Notably, the trend of the percentages of Sox2-EGFP positive cells was decreasing while the trend of that in Myosin VIIa positive hair cells was increasing during 3-day to 14-day post-5-aza injection (Figure 2B). In addition, the percentages of double-labeled cells reached to a peak at five days following surgery, which was the time point when the percentages of supporting cells and hair cells turned stable (Figure 2B). The number of coexpressed cells fell to zero at two weeks post-surgery (Figure 2B), indicating the completion of the transition from supporting cells to new hair cells. These data suggest that the transition of hair cells from supporting cells may occur approximately during 5–7 days after 5-aza treatment. In the meantime, Myosin VIIa-expressing cells were not detected in the control group of deafened mice treated with saline (Figure 3), indicating the absence of regenerated hair cells two weeks following saline treatment. Taken together, the data of this study indicate that regeneration of hair cells by 5-aza may arise from Sox2 positive supporting cells during 5–7 days after 5-aza treatment.

## 5. Conclusions

In summary, this research investigates the cellular mechanisms to regenerate hair cells in adult mice via DNA demethylation. The results reveal that regenerated hair cells by 5-aza injection may come from Sox2-EGFP positive supporting cells. However, there are several unaddressed questions in our endeavor, which are too complicated to include in a single study but will be investigated as independent studies in the future. For instance, it is unclear whether the transition of supporting cells to hair cells will cause a deficient number of supporting cells or whether reduced supporting cells can be compensated by the cell cycle reentry of supporting cells. Additionally, the function of the regenerated hair cells and the molecular mechanisms of this method are yet to be determined. Taken together, this study investigates the cell source of hair cell regeneration via an epigenetic approach. The outcomes of this study may shed light on the mechanisms important for the regeneration of cochlear hair cells. The advantage of the epigenetic method is to regulate gene expression without changing the DNA sequence, which is an important aspect for future clinical applications.

## Figures and Tables

**Figure 1 ijms-21-08649-f001:**
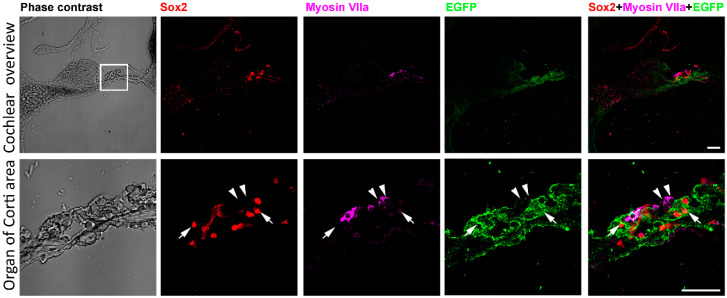
Expression of Sox2/EGFP in the cochlea. Low magnification phase contrast figures show cochlear section overview, Sox2 immunostaining, Myosin VIIa immunostaining, EGFP expression and the merged image of Sox2, Myosin VIIa and EGFP. The box area indicates the organ of Corti and the high magnification images were shown in the second row. High magnification images demonstrate the organ of Corti area showing the phase contrast overview, Sox2 immunostaining (arrows), Myosin VIIa immunostaining (arrowheads), EGFP expression (arrows) and the merge image of Sox2, Myosin VIIa and EGFP (arrows and arrowheads). It is shown that hair cells are labeled by Myosin VIIa antibodies (arrowheads). Surrounding supporting cells express EGFP and are labeled by Sox2 antibodies (arrows). Scale bar: 50 µm.

**Figure 2 ijms-21-08649-f002:**
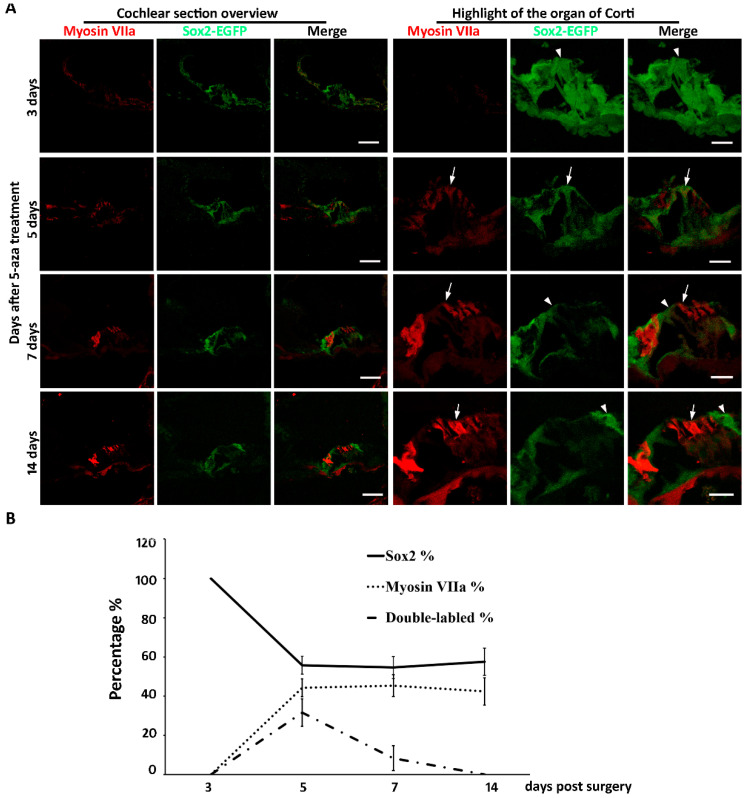
(**A**) The fate of supporting cells and hair cells after 5-aza injection. Three days following 5-aza treatment, Sox2-EGFP-expressing cells are observed (arrowhead), but they lack Myosin VIIa expression. Five days following 5-aza treatment, weak Myosin VIIa immunostaining is observed; some cells express both Myosin VIIa and Sox2-EGFP (arrow). During 7–14 days after treatment, cells in the organ of Corti express only Myosin VIIa (arrow) or Sox2-EGFP (arrowhead). Scale bar: 50 µm in cochlear section overview; 20 µm in organ of Corti highlight. (**B**) Quantitative study shows the percentages of Sox2-EGFP positive supporting cells, Myosin VIIa expressing hair cells and double labeled cells from 3-day to 14-day post-5-aza treatment.

**Figure 3 ijms-21-08649-f003:**
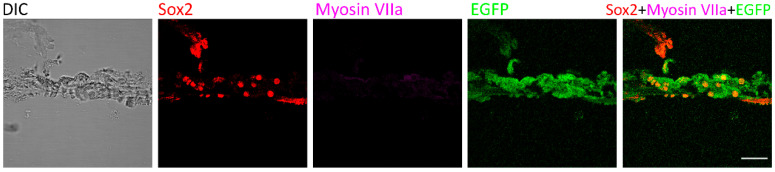
Myosin VIIa expression was not detected in the control mice. Two weeks after saline treatment, the expression of Myosin VIIa was not detected in the control group that was treated with saline following deafening. Sox2 and EGFP were observed in the organ of Corti. Scale bar: 50 µm.

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
