# Peer review of "Generation of Cochlear Hair Cells from Sox2 Positive Supporting Cells via DNA Demethylation"

_ijms, 2020, doi:10.3390/ijms21228649_

Round 1

Reviewer 1 Report

The paper is interesting in that it provides useful and additional information to improve the approach to gene therapy

Author Response

The paper is interesting in that it provides useful and additional information to improve the approach to gene therapy”.

Response: We thank the reviewer for the helpful comments.

Reviewer 2 Report

This manuscript has limited evidence to prove the conclusion.

The conclusion of this study was solely based on epifluorescence analysis of sectioned image of organ of Corti and representative image shown in fig2 is not highly specific.

Functional analysis such as ABR, or DPOAE and frequency specific co-expression of GFP and Myosin VIIa should be demonstrated by whole mount preparation with better quantification. Statistical analysis is necessary to prove such scientific observation which lacks in this study.

Furthermore, based on the fact that progenitor cells which could differentiated into cochlear hair cells are known to progressively disappear after birth, theory that pluripotent stem cells or progenitor cells would differentiate into the mature hair cell in specific location by otic differentiation enhancer in adult mouse is not appropriate. 

Author Response

Reviewer #2

Point 1:Functional analysis such as ABR, or DPOAE and frequency specific co-expression of GFP and Myosin VIIa should be demonstrated by whole mount preparation with better quantification.”

Response: We thank the reviewer for the comments. A detailed investigation of functional evaluation including pure-tone ABR and DPOAE is in process, which may deserve as an independent study (line 264). Statistical analysis of the quantification in Fig. 2 has been included (line 115 and 180).

Point 2:Furthermore, based on the fact that progenitor cells which could differentiated into cochlear hair cells are known to progressively disappear after birth, theory that pluripotent stem cells or progenitor cells would differentiate into the mature hair cell in specific location by otic differentiation enhancer in adult mouse is not appropriate.”

Response: Indeed, mammalian cochlear hair cell progenitors lose the differentiation ability shortly after birth. The reason is still obscure. In in this study, a novel epigenetic approach is used to address this issue. The outcomes of this study may shed light on the mechanisms important for the regeneration of cochlear hair cells (line 267). 

Reviewer 3 Report

Dear Author,

The article is focused on  an interesting and original research, though a very specific topic has been analized. Since it was already known that Sox2 gene play a basic role in the development of the inner ear, this attempt to regenerate HCs starting from SCs is hopeful for the future studies.

According to me, You can improved these parts : -the structure of the article: the paragraph"4-Materials and Methods" will be more useful if following the introduction and the aim of this paper, in order to understand the scientific procedure before coming to the Results and Discussion; -the control group has been observed for 14 days, it's not clear if it has gone through the cochlear sectioning such as the other groups, -the text is full of abbreviations and acronyms  that make reading not very fluid, -Fig. 2B could more attractive, but is already a good explanation to the trends of the percentages of Myosin VIIa and Sox2  -the use of 5-azacytidine caused side effects in the mice?  thinking about a proper use in the future of this chemotherapy

Author Response

Reviewer #3:

Point 1:the structure of the article: the paragraph"4-Materials and Methods" will be more useful if following the introduction and the aim of this paper, in order to understand the scientific procedure before coming to the Results and Discussion”.

Response: We thank the reviewer for this valuable comment. The "Materials and Methods" section has been moved according to the suggestion (line 68). 

Point 2:the control group has been observed for 14 days, it's not clear if it has gone through the cochlear sectioning such as the other groups”.

Response: The sectioning and immunostaining of the control mice was performed in the same way as the 5-aza groups. We have added the details in the text (lines 93 and 96).”

Point 3:the text is full of abbreviations and acronyms that make reading not very fluid”.

Response: We have reduced the abbreviations and revised the text to make reading fluid. For instance, we have changed “hc” to “hair cell”.

Point 4:Fig. 2B could more attractive, but is already a good explanation to the trends of the percentages of Myosin VIIa and Sox2”. 

Response: We have added statistical analysis of Fig. 2B (line 115 and 180).

Point 5:if the use of 5-azacytidine caused side effects in the mice?” 

Response: The side effects issue has been addressed in our previous publication (Deng et al 2019, Scientific Reports). Briefly, 4 mM 5-aza was injected into the mouse inner ear. No inflammation or other significant side effect was observed in all injected mice for up to six weeks following injection. Therefore, 5-aza is likely to be safe if used in a proper concentration in the future application as a chemotherapy. We had added this in the discussion section (line 228).

Round 2

Reviewer 2 Report

It is very difficult to agree on publishing this manuscript in current form without any functional evidence or confirmative wholemount images to clearly identify the co-expressions. 

I recommend authors to do additional experiment to prove this in wholemount preparation with showing the possibility that supporting cells with LGR5 (known progenitor factor) expressions have been differentiated into hair cells. 

Author Response

Reviewer #2 “recommend authors to do additional experiment to prove this in wholemount preparation with showing the possibility that supporting cells with LGR5 (known progenitor factor) expressions have been differentiated into hair cells.”

Response:  We appreciate the reviewer for his/her helpful comments in terms of the Lgr5 mouse and wholemount preparation. We apologize that we have not made it very clear why the Sox2-EGFP mouse and the cochlear section approach were selected in this study. We would like to take this opportunity to explain it and we have included these explanations to the revised manuscript.  

Indeed, Lgr5-expressing supporting cells are able to self-renew and give rise to hair cells. Meanwhile Lgr5 is only expressed in a subset of supporting cells of the adult murine cochlea (Shi et al., 2012). The major aim of this study is to investigate whether supporting cells are the cell source for newly generated hair cells. To identify all types of supporting cells, it is a logic step to pre-label the cochlear sections with Sox2 in the current stage of this project, because Sox2 is known to express in all types of supporting cells in the mature murine inner ear (Hume et al., 2007). Therefore, the Sox2-EGFP mouse is an appropriate animal model for the current study and was selected in this research (line 40&63).

As for wholemount vs. cochlear section, we have shown newly generated hair cells using the wholemount preparation in our previous publication (Deng et al., 2019). In the organ of Corti, hair cells locate above the supporting cells. The wholemount preparation can observe surface cells including hair cells and some supporting cells, whereas the supporting cells beneath the surface hair cells are not ready to detect easily. The major aim of this study is to investigate whether new hair cells are generated from supporting cells. It is critical to visualize both hair cells and supporting cells simultaneously. The cochlear section approach provides a side view of the organ of Corti, which is able to observe hair cells and supporting cells in the sections. Therefore, the cochlear section is a suitable approach and was selected in this study (line 97).

Taken together, the Sox2-EGFP mouse model and the cochlear section approach are selected to achieve the research aim of this study. We will appreciate that the reviewer will kindly accept our selection.

Round 3

Reviewer 2 Report

No further comment.